# Ablation of DNA-methyltransferase 3A in skeletal muscle does not affect energy metabolism or exercise capacity

**Lewin Small**[1], **Lars R. Ingerslev**[1], **Eleonora Manitta**[1], **Rhianna C. Laker**[1], **Ann N. Hansen**[1], **Brendan Deeney**[1], **Alain Carrié**[2], **Philippe Couvert**[2], **Romain Barrès**[1]*

**1** Novo Nordisk Foundation Center for Basic Metabolic Research, Faculty of Health and Medical Sciences, University of Copenhagen, Copenhagen, Denmark, **2** Sorbonne Université-INSERM UMR_S 1166 ICAN, Pitié-Salpêtrière Hospital, Paris, France

* barres@sund.ku.dk

## Abstract

In response to physical exercise and diet, skeletal muscle adapts to energetic demands through large transcriptional changes. This remodelling is associated with changes in skeletal muscle DNA methylation which may participate in the metabolic adaptation to extracellular stimuli. Yet, the mechanisms by which muscle-borne DNA methylation machinery responds to diet and exercise and impacts muscle function are unknown. Here, we investigated the function of *de novo* DNA methylation in fully differentiated skeletal muscle. We generated muscle-specific DNA methyltransferase 3A (DNMT3A) knockout mice (mD3AKO) and investigated the impact of DNMT3A ablation on skeletal muscle DNA methylation, exercise capacity and energy metabolism. Loss of DNMT3A reduced DNA methylation in skeletal muscle over multiple genomic contexts and altered the transcription of genes known to be influenced by DNA methylation, but did not affect exercise capacity and whole-body energy metabolism compared to wild type mice. Loss of DNMT3A did not alter skeletal muscle mitochondrial function or the transcriptional response to exercise however did influence the expression of genes involved in muscle development. These data suggest that DNMT3A does not have a large role in the function of mature skeletal muscle although a role in muscle development and differentiation is likely.

## Author summary

Skeletal muscle is a plastic tissue able to adapt to environmental stimuli such as exercise and diet in order to respond to energetic demand. One of the ways in which skeletal muscle can rapidly react to these stimuli is DNA methylation. This is when chemical groups are attached to DNA, potentially influencing the transcription of genes. We investigated the function of DNA methylation in skeletal muscle by generating mice that lacked one of the main enzymes responsible for *de novo* DNA methylation, DNA methyltransferase 3A (DNMT3A), specifically in muscle. We found that loss of DNMT3A reduced DNA methylation in muscle however this did not lead to differences in exercise capacity or energy

**Data Availability Statement:** The authors confirm that all data underlying the findings are fully available without restriction. Transcriptomic and DNA methylation data have been deposited in a

publicly available database (https://www.ncbi.nlm.
nih.gov/geo/query/acc.cgi?acc=GSE152349) and
numerical data underlying all figures is provided in
a spreadsheet in the supporting information (S2
Spreadsheet).

**Funding:** The Novo Nordisk Foundation Center for
Basic Metabolic Research is an independent
research center at the University of Copenhagen,
partially funded by an unrestricted donation from
the Novo Nordisk Foundation (NNF18CC0034900).
L.S. is supported by a research grant from the
Danish Diabetes Academy, which is funded by the
Novo Nordisk Foundation (NNF17SA0031406). The
funders had no role in study design, data collection
and analysis, decision to publish, or preparation of
the manuscript.

**Competing interests:** The authors have declared
that no competing interests exist.

metabolism. This suggests that DNMT3a is not involved in the adaptation of skeletal muscle to diet or exercise.

## Introduction

Regulation of gene expression is a critical process in all cells and plays a defining role in achieving tissue- and cell-specificity, in the face of an identical genetic code. The control of gene expression is dependent upon transcription factors which, upon activation, bind to transcription factor-specific DNA motifs in promoter and/or enhancer regions, recruit co-factors and initiate transcription. Epigenetic modifications to DNA and histones add an additional level of control to gene expression by altering the structure of DNA and subsequent accessibility of transcriptional machinery to gene regulatory elements. DNA methylation is a major epigenetic modification that occurs at the 5' position of cytosine residues within a cytosine-guanine (CpG) dinucleotide resulting in 5-methylcytosine. The presence or absence of 5-methylcytosine modulates the recruitment of methyl binding domain (MBD) proteins to the DNA and depending on the genomic context, may regulate transcription factor binding [1].

Skeletal muscle is a highly plastic tissue that displays a robust adaptive response to environmental stimuli such as physical exercise and diet. Muscle contraction results in large-scale transcriptional remodelling of metabolic, antioxidant and contractile genes which participate in the adaptation to the increased demands placed on the tissue [2]. This transcriptional remodelling is associated with changes in DNA methylation after both acute exercise [3–5] and exercise training [5–7]. For example, intense exercise in humans (80% maximal aerobic capacity) resulted in an immediate reduction in promoter methylation of *PPARGC1A*, *TFAM*, *MEF2A* and *PDK4* and expression levels of these genes were upregulated 3 hours later in an intensity-dependent manner [3]. However, exercise-mediated demethylation of promoter regions is transient and re-methylation quickly occurs, within 1–3 hours [3], suggesting possible directed methylation. Changes in nutritional status, either through diabetes status [8], fasting [9], weight loss [10] or diet [7,11] have also been shown to alter DNA methylation in skeletal muscle.

Despite an abundance of evidence that environmental stimuli such as exercise and diet can affect DNA methylation in skeletal muscle [12,13], the mechanisms by which DNA methylation machinery responds to these stimuli are still unclear. The regulation of DNA methylation is complex and involves multiple enzymes that actively methylate (DNA methyltransferases, DNMTs) or oxidize for potential demethylation (Ten-eleven translocation enzymes, TETs) of cytosine residues. DNA may also be demethylated passively (non-enzymatically) during cell division. The classical model of DNA methylation proposes that DNMT1 is responsible for maintaining DNA methylation marks through cell divisions, while DNMT3A and DNMT3B are thought to be primarily responsible for *de novo* methylation events [14]. Early work in the field observed that knockdown of *DNMT3B* in muscle cells *in vitro* could impede fatty acid-induced methylation of the *PPARGC1A* promoter [8]. However, as *Dnmt3a* is the most highly expressed DNMT isoform in skeletal muscle [15] we decided to focus on the role of DNMT3A in differentiated skeletal muscle to investigate the importance of *de novo* DNA methylation in the response of skeletal muscle to environmental stimuli (exercise and diet).

We generated muscle-specific DNMT3A knockout mice (mD3AKO) and investigated the impact of the loss of muscle DNMT3A on skeletal muscle DNA methylation, exercise capacity, energy metabolism and mitochondrial function. Additionally, by integrating DNA

methylation and transcriptomics data, we report novel DNMT3A targets in differentiated skeletal muscle, regulated by promoter/gene body methylation.

## Results

### mD3AKO mice have reduced skeletal muscle DNA methylation

In order to delete *Dnmt3a* in muscle, we utilized the Cre-lox system with a muscle creatine kinase promoter driving Cre recombinase expression. mD3AKO mice had substantially reduced DNMT3A at the protein level compared to WT mice, specifically in skeletal muscle and heart and not in other tissues such as liver, kidney or brain (Fig 1A and 1B). DNA methylation analysis and RNA sequencing were performed on the *soleus* muscle as this muscle displays dynamic de-methylation and then re-methylation of metabolically important genes following contraction [3]. mRNA levels of *Dnmt3a* showed a ~60% reduction in the *soleus*, EDL and *quadriceps* muscles of mD3AKO mice compared to wild type (WT) mice (Figs 1C and S1A). As mCK-Cre is expressed quite late during skeletal muscle differentiation [16], the residual *Dnmt3a* expression likely comes from satellite cells and non-muscle cells (immune cells, fibroblasts, adipocytes). mRNA levels of *Dnmt3b*, the other major *de novo* DNA methyltransferase isoform, showed no compensatory upregulation in mD3AKO muscle and was expressed at a much lower level than *Dnmt3a* (Fig 1C). To quantify DNA methylation, we performed reduced representation bisulfite sequencing (RRBS) on the *soleus* muscle of WT and mD3AKO mice. There was a small but significant reduction in global DNA methylation (percentage of quantified CpG sites that were methylated) in the muscle of mD3AKO animals (Fig 1D). This corresponded to reduced methylation in mD3AKO muscle compared to WT in all genomic contexts examined (Fig 1E) however, the largest fold changes were in CpG islands (CGI, Fig 1F). After correcting for multiple testing (False Discovery Rate, FDR 5%), we found 1422 differentially methylated clusters between WT and mD3AKO muscle. 1386 were hypomethylated and 36 were hypermethylated. Methylation at 9655 promoter regions was quantified, 184 promoter regions were significantly hypomethylated and 13 promoter regions were significantly hypermethylated in mD3AKO muscle compared to WT (Fig 1G). Gene ontology analysis (gene ontology—biological process) of hypomethylated clusters included significantly enriched terms related to muscle development (S1B Fig), while there was no significant enrichment of terms from hypermethylated regions. Gene ontology analysis of hypomethylated promoter regions revealed only two terms: "inhibition of cysteine-type endopeptidase activity" and "zymogen inhibition" (S1C Fig).

### DNMT3A knockout in skeletal muscle does not affect exercise capacity or the transcriptional response to acute exercise

In order to assess exercise capacity in mD3AKO mice, we investigated both the capacity for these mice to run voluntarily, with free access to running wheels, as well as during a forced treadmill exercise tolerance test. For both exercise regimens, we observed that males ran a shorter distance compared to females. mD3AKO mice with free access to a running wheel ran a similar distance and speed compared to WT mice (Fig 2A and 2B). Similarly, there was no genotype-specific differences in the distance run during forced exercise (Fig 2C). Blood lactate, a measure of glucose utilization during exercise, showed an expected increase after exercise however there was no effect of genotype (Fig 2D).

In a separate cohort of 12-week-old mice, we assessed the transcriptional response to exercise in WT and mD3AKO mice by harvesting *soleus* muscle 60 minutes following a 30-minute treadmill exercise bout comparing it to muscle from non-exercised mice (Fig 2E). We found 1615

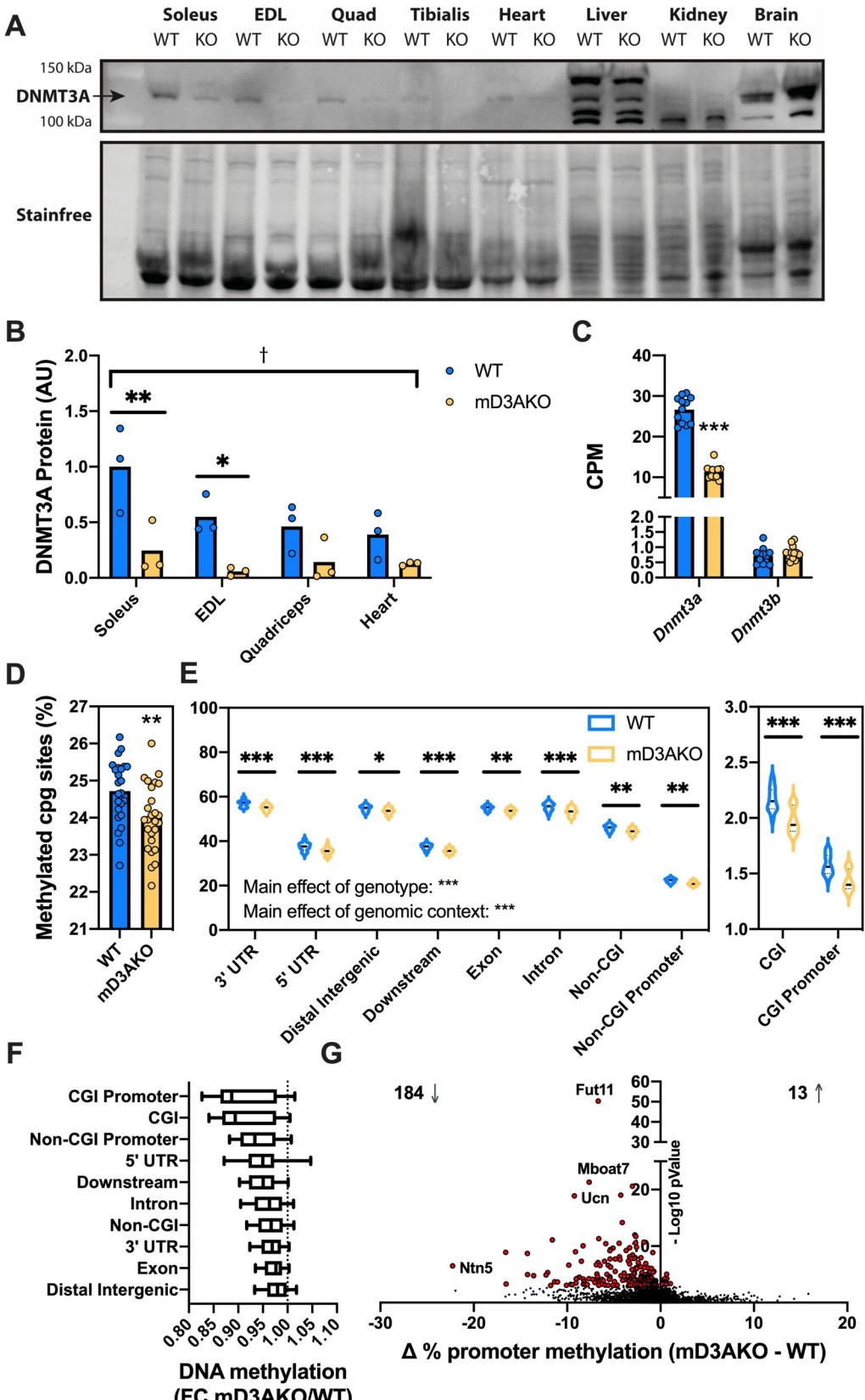

**Fig 1. Ablation of DNMT3A in muscle reduces DNA methylation over multiple genomic contexts.** (A) Western blot of DNMT3A protein in multiple tissues from WT and mD3AKO mice and (B) quantitation of DNMT3A protein in skeletal muscle, analysed by 2-way ANOVA, n = 3. (C) *Dnmt3* isoform mRNA expression (quantified by RNA-seq) in the *soleus* muscle analysed by individual t-tests, n = 12. (D) Percentage of total methylated cytosine residues quantified by RRBS in *soleus* muscle, analysed by t-test and (E) Quantified CpG methylation in different genomic contexts, analysed by 2-way ANOVA. (F) Fold-change of methylated cytosine residues comparing mD3AKO to WT muscle in different genomic contexts. (G) Volcano plot of differentially methylated promoter regions, red points are significant, n = 23–24 mice. * P < 0.05, ** P < 0.005, *** P < 0.0005, † P < 0.0005 main effect of genotype. Data show the mean (bar) with individual data points or violin plots.

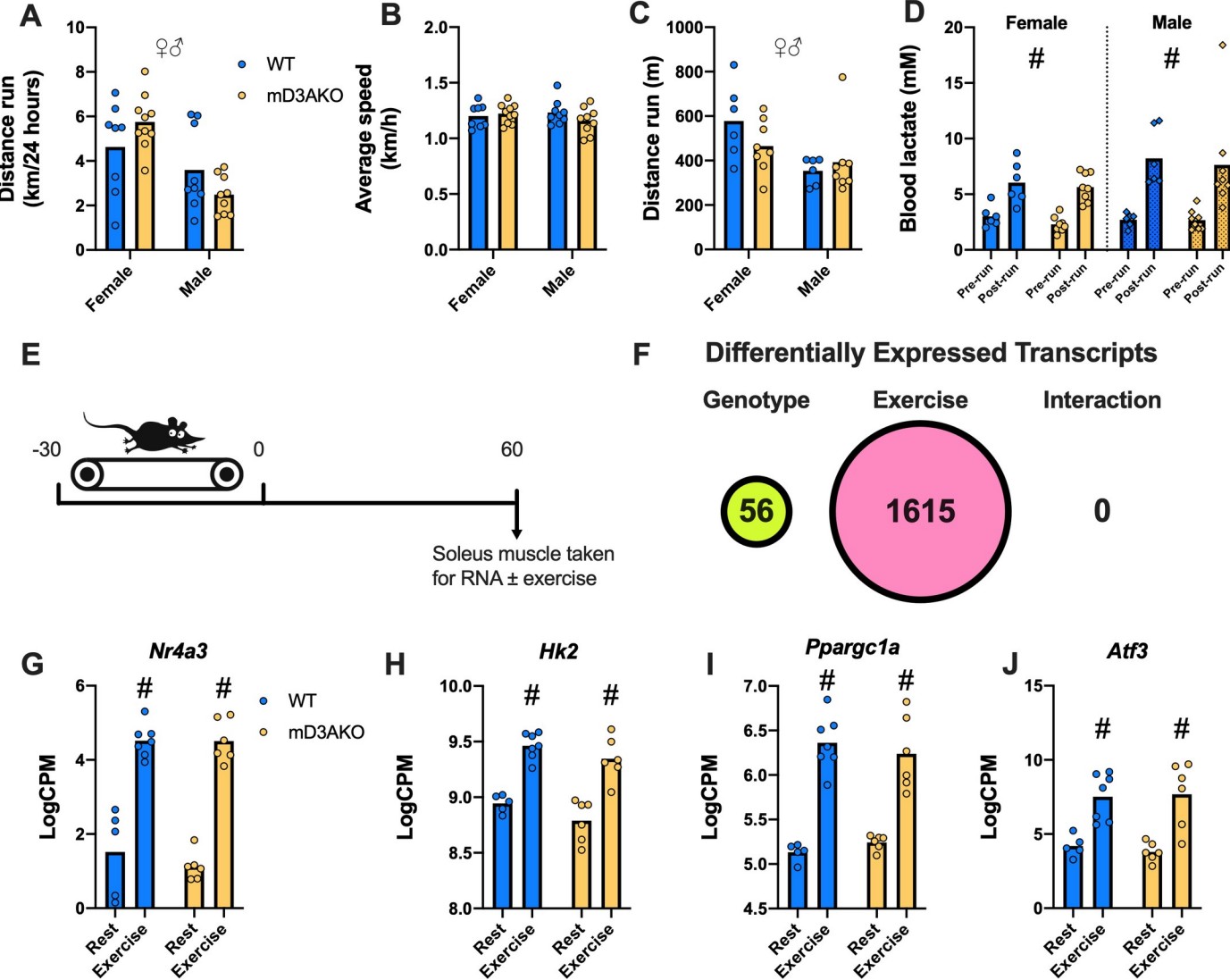

**Fig 2. Ablation of DNMT3A in skeletal muscle does not affect exercise capacity or the transcriptional response to acute exercise.** In a cohort of 32-36-week-old chow-fed mice, (A) voluntary wheel running distance and (B) speed in mice averaged for a 24-hour period from a period of 2 weeks, n = 8–10. (C) Distance run during a graded exercise tolerance test and (D) blood lactate before and directly after test, n = 6–8. Analysed by 2-way ANOVA for a main effect of genotype and a main effect of sex (A-C) or a main effect of genotype and a main effect of exercise, D. (E) Simplified schematic of acute running treadmill experiment in a separate cohort of 12-week old mice. (F) Number of significantly differentially expressed genes in the *soleus* muscle when comparing genotype, exercise and a genotype/exercise interaction. Expression of exercise responsive genes (G) *Nr4a3*, (H) *Hk2*, (I) *Ppargc1a* and (J) *Atf3* in the *soleus* muscle of combined male and female mice, n = 5–7. ♀♂ P < 0.05 main effect of sex, # P < 0.05 main effect of exercise. Data show the mean (bar) with individual data points.

genes with a significant change in expression between the exercised and sedentary muscles (FDR 5%), while 56 were altered by genotype. However, there were no genes that had a significant exercise/genotype interaction (Fig 2F). Well-characterized exercise-responsive genes such as *Nr4a3*, *Ppargc1a* and *Hk2* (Fig 2G–2I) showed robust increases in expression after exercise but here again, we found no interaction between exercise and genotype. Activating transcription factor 3 (*Atf3*) displayed the largest increase in expression after exercise (~15 fold increased, Fig 2J).

Because few differences were found in exercise capacity or the transcriptional response to exercise between WT and mD3AKO mice, we used the methylation and RNA-seq data generated from *soleus* muscle to investigate the interaction between methylation and mRNA changes in response to genotype and exercise. When the changes in muscle DNA methylation were visualized by principal component analysis (PCA), a technique used to separate multi-dimensional data based on variance, there was a clear separation of the muscle methylation data by genotype (Fig 3A), however no clear separation by exercise (Fig 3B), suggesting that ablation of DNMT3A has a far greater effect on methylation in muscle than exercise. This was also evident when looking at the number of differentially methylated regions when comparing the effect of genotype to the effect of time after exercise (S2A Fig). Interestingly, the RNA-seq data showed the opposite result, with multidimensional scaling (MDS) plots highlighting a far greater separation by exercise (Fig 3D) than genotype (Fig 3C). A separation by sex from the RNA-seq data was also apparent (Fig 3C). Global DNA methylation after exercise was not altered by time after exercise however, there was a clear reduction in DNA methylation in the muscle of mD3AKO mice (S2B Fig).

## mD3AKO mice have normal body composition, whole-body energy expenditure, substrate utilization and glucose tolerance

To investigate if DNMT3A in skeletal muscle contributes to whole-body energy metabolism, we measured body composition, whole-body energy expenditure (EE) and glucose tolerance in 30-week-old WT and mD3AKO mice fed a standard chow diet. mD3AKO had a similar body weight, lean mass and fat mass compared to WT mice (Fig 4A–4D). We observed no difference in substrate utilization (RER, Fig 4E and 4F) or EE (Fig 4G and 4H) between WT and mD3AKO mice, although these measurements displayed clear diurnal variations, as expected. Male mice had a significantly higher rate of EE than female mice (Fig 4G and 4H) and there was a positive correlation between EE and lean mass regardless of sex and diet however this was not altered by genotype (Fig 4I). Glucose tolerance, fasting blood glucose, plasma insulin and NEFA levels were not affected by genotype (Fig 4J–4N) however, fasting plasma insulin levels were substantially higher in male mice (Fig 4M). Muscle mass, myofiber size and fiber composition (determined by analysis of gene transcription of slow and fast twitch muscle markers) were all similar between WT and mD3AKO mice (S3 Fig).

## mD3AKO mice have a normal response to high-fat feeding

We next wanted to determine if skeletal muscle DNMT3A had a role in high fat diet (HFD)-induced glucose intolerance or adiposity as a previous investigation found that DNMT3A ablation in adipose tissue improved glucose tolerance in mice fed a HFD but not in chow-fed mice [17]. Male WT or mD3AKO mice were fed a HFD (45% calories from fat) for a period of 12 weeks after which body composition was analysed and a glucose tolerance test was performed. Indirect calorimetry was performed 2 weeks later (14 weeks of diet, Fig 5A). HFD-fed mice had substantially increased body and fat mass compared to chow-fed mice however there was no effect of genotype (Fig 5B and 5C). Similarly, there was no difference in glucose tolerance between HFD-fed mD3AKO and WT mice, although HFD-fed mice had a higher glucose excursion than chow-fed mice indicating relative glucose intolerance (Fig 5D and 5E). The

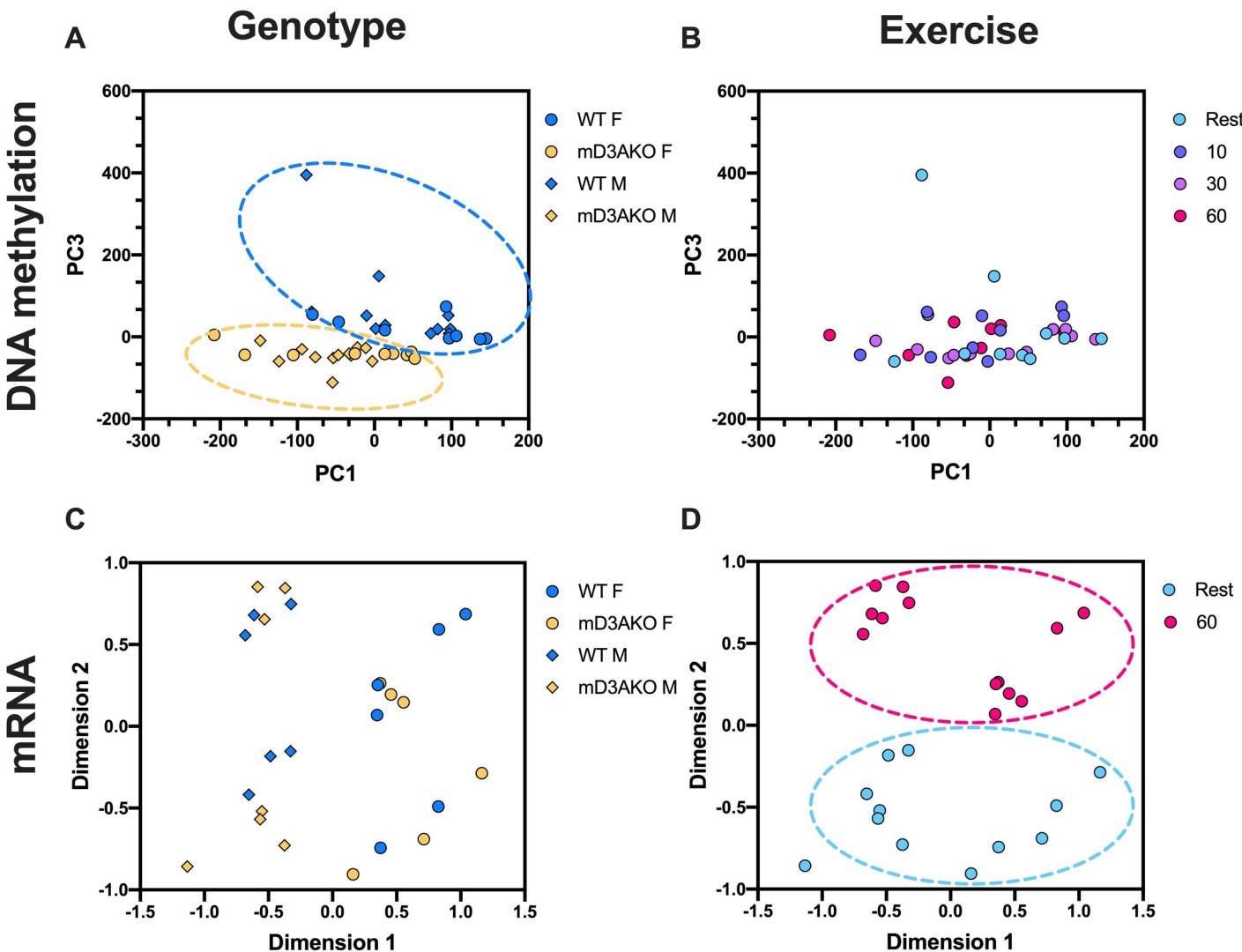

**Fig 3. Genotype has a stronger effect on skeletal muscle DNA methylation than exercise while the opposite is true of skeletal muscle mRNA abundance.** PCA of DNA methylation data from RRBS labelled for (A) genotype and (B) time after exercise. MDS plots of RNA-seq data labelled for (C) genotype and (D) time after exercise. From *soleus* muscle of 12-week old male and female WT and mD3AKO mice sacrificed 10, 30 or 60 minutes after 30 minutes of treadmill running, or not exercised (rest).

whole-body EE and RER of HFD-fed mD3AKO mice were similar to WT mice (Fig 5F and 5G). Independent of genotype, HFD-feeding caused an expected decrease in 24-hour RER compared to chow-fed mice indicating a greater reliance on fat oxidation (Figs 4F and 5G). Additionally, we analysed *Dnmt3a* expression in the muscle of WT mice fed either chow or a HFD. 16–18 weeks of high-fat feeding did not alter the expression of *Dnmt3a* in *soleus* or *quadriceps* muscle compared to age-matched chow-fed mice however, *Dnmt3a* expression was increased ~1.4 fold in the EDL of HFD-fed mice (Fig 5H).

## Loss of DNMT3A in muscle does not affect mitochondrial content or function

As skeletal muscle is responsible for a substantial proportion of whole-body energy utilization [18] and DNMT3A has been reported to be present in the mitochondria [19] we

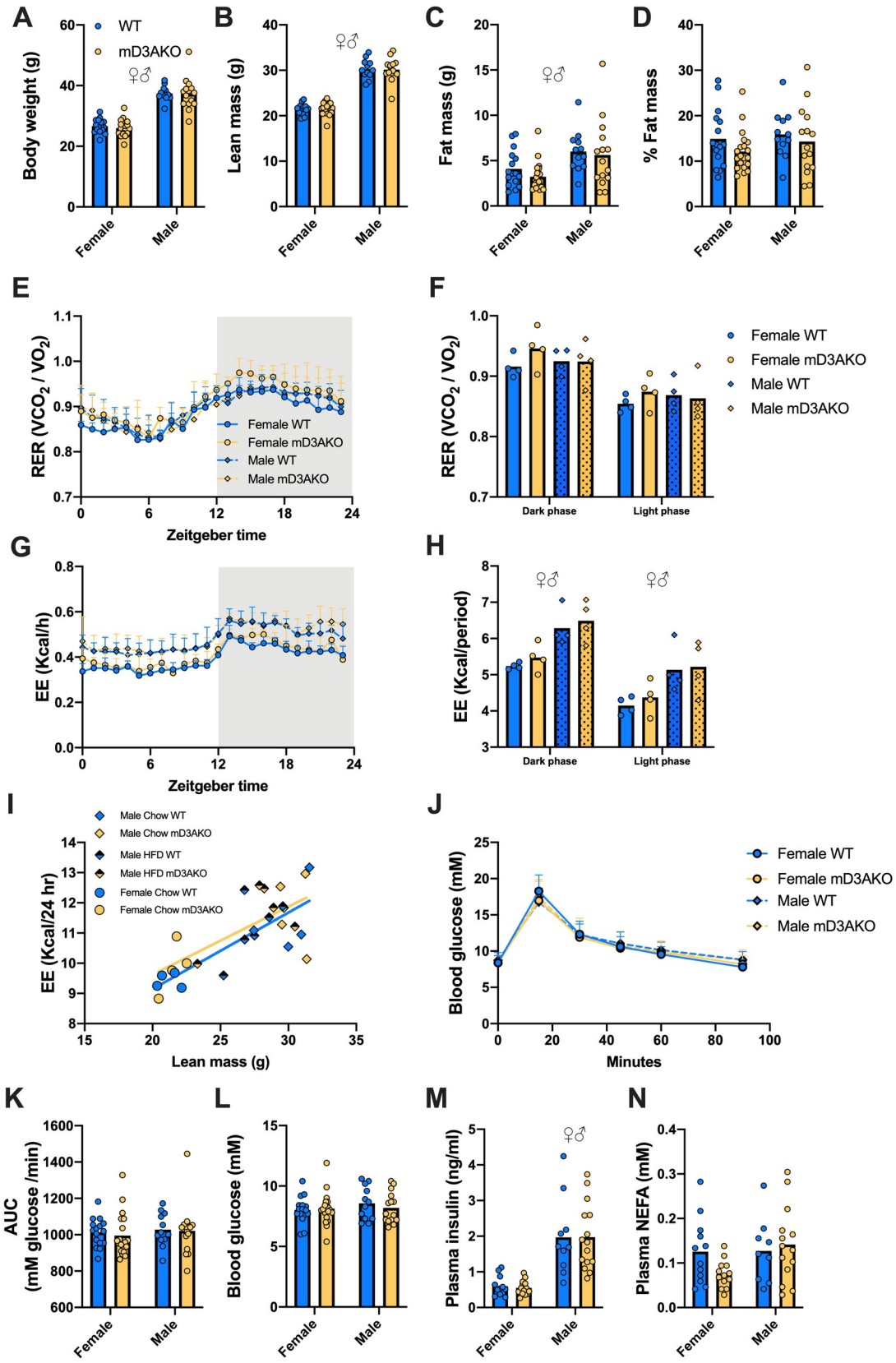

**Fig 4. mD3AKO mice have normal body composition, glucose tolerance and energy expenditure.** (A) Body weight, (B) lean mass, (C) fat mass and (D) percentage of body mass in chow-fed 30-week-old male and female WT and mD3AKO mice, n = 12–20. (E) Diurnal respiratory exchange ratio and (F) average during the light and dark phase. (G) Diurnal energy expenditure and (H) total during the light and dark phase, measured in 32-36-week-old chow-fed male and female WT and mD3AKO mice, n = 4. (I) ANCOVA comparing the relationship between lean mass and energy expenditure in WT and mD3AKO mice combining sex and diet, n = 14. (J) Oral glucose tolerance test curves, (K) the area under the curve of the oGTT curves and (L) blood glucose after a 6 hour fast in chow-fed 30-week-old male and female WT and mD3AKO mice. (M) Plasma insulin and (N) plasma NEFA from blood taken after sacrifice in 6-hour fasted 36-40-week-old mice. n = 9–15. Analysed by 2-way ANOVA for a main effect of genotype and a main effect of sex. ♀♂ P < 0.05 main effect of sex. Data show the mean (bar) with individual data points or mean +SD.

investigated if DNMT3A is required for proper mitochondrial function. The *soleus* and EDL from mD3AKO mice had similar abundances of OXPHOS complexes to WT muscle, although EDL from male mice had a significant increase in complex III abundance compared to female mice (Fig 6A–6C). Similarly, *soleus* mtDNA copy number, a measure of mitochondrial content, was not different between genotypes although male mice displayed an increase in *soleus* mtDNA copy number compared to females (Fig 6D). Next, we isolated muscle satellite cells from male mD3AKO and WT mice to investigate mitochondrial function *in vitro*. mD3AKO-derived primary myotubes had a significant (70%) reduction in mRNA levels of *Dnmt3a* only at day 5 of differentiation (harvested for RNA in parallel with the Seahorse experiments) (Fig 6E). Compared to myotubes derived from WT mice, 5-day differentiated primary myotubes from mD3AKO mice showed no differences in basal or FCCP-uncoupled oxygen consumption (Fig 6F) or the extracellular acidification rate (Fig 6G) during Seahorse experiments.

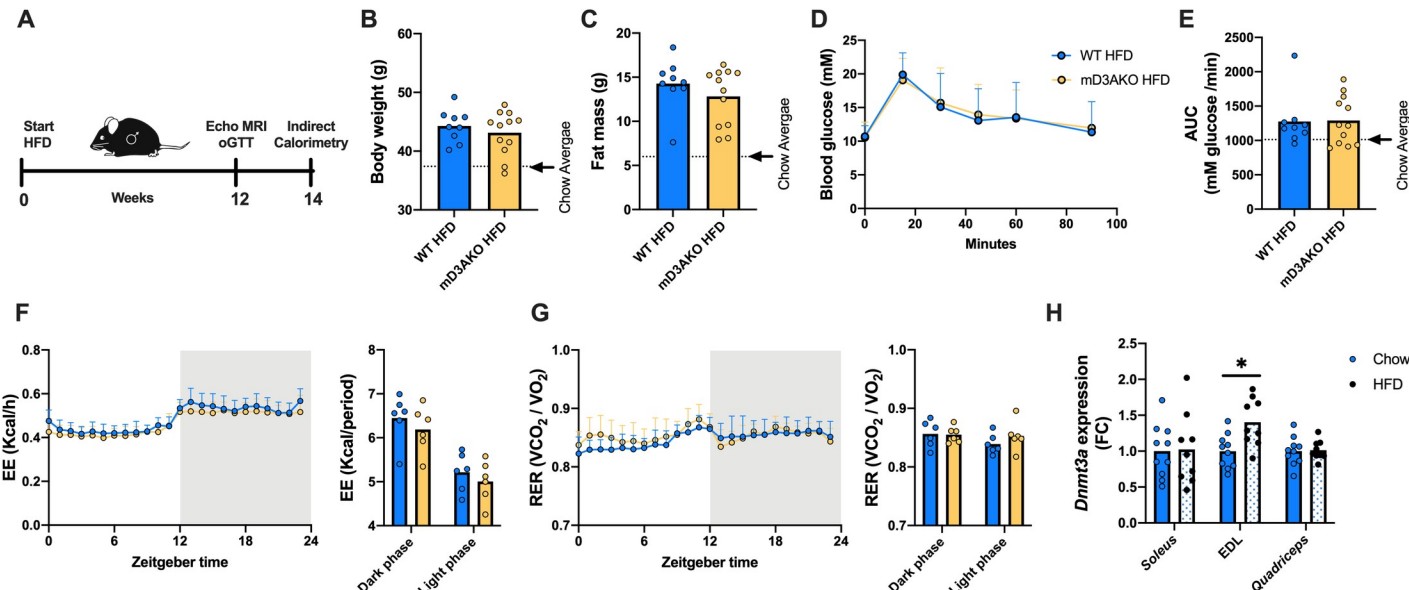

**Fig 5. Ablation of DNMT3A in skeletal muscle does not affect the whole-body response to high-fat feeding.** (A) 18-week-old male WT and mD3AKO mice were fed a HFD for a period of 12 weeks after which body composition and glucose tolerance was determined. Indirect calorimetry was performed at 14 weeks of HFD. (B) Body weight, (C) fat mass, (D) oral glucose tolerance test curves and (E) the area under the curve of the oGTT curves. The mean of the chow values is depicted by the dotted line, n = 9–12. (F) Respiratory exchange ratio and (G) energy expenditure, n = 6. (H) *Dnmt3a* expression in the *soleus*, EDL and *quadriceps* muscles of WT chow and HFD-fed mice compared to the housekeeping gene *18S*, fold-change compared to WT, n = 9–10. Analysed by individual t-tests. * P < 0.05. Data show the mean (bar) with individual data points or mean +SD.

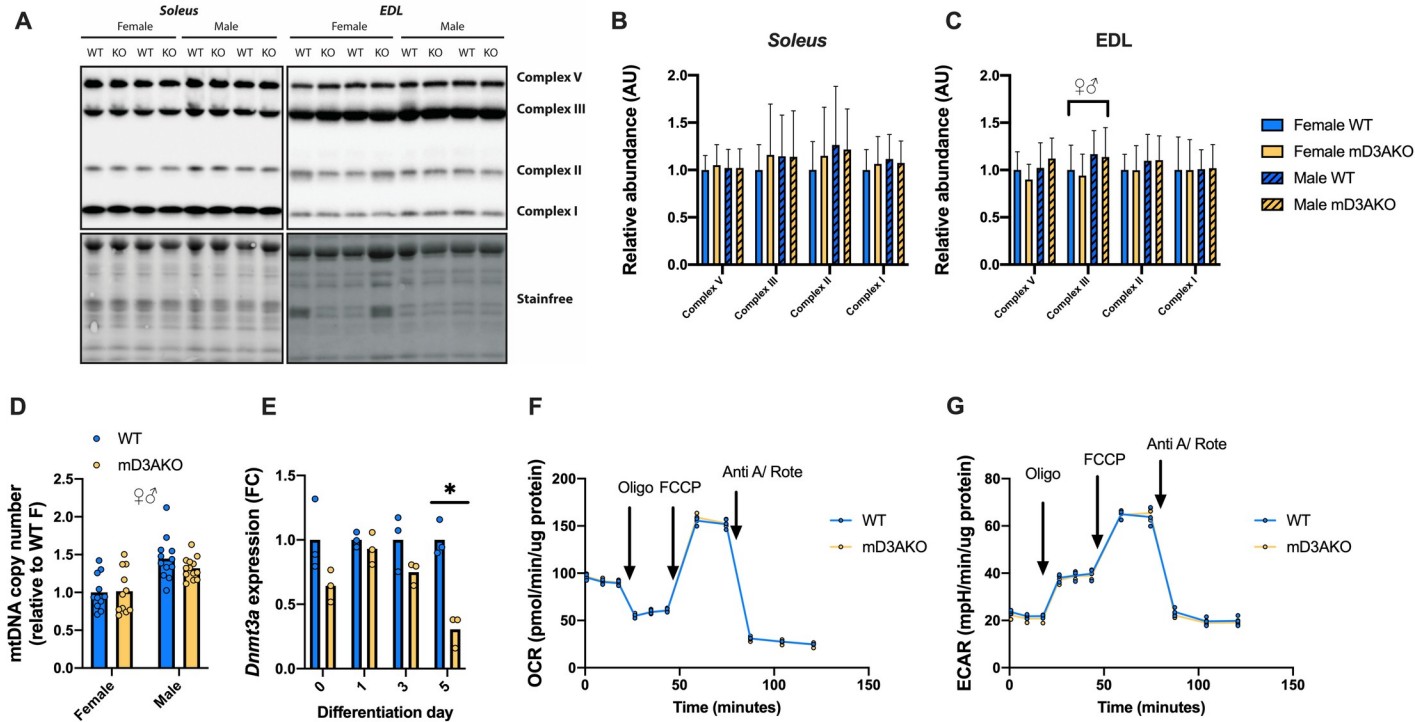

**Fig 6. Loss of DNMT3A does not affect skeletal muscle mitochondrial function.** (A) Representative blot and densitometry of OXPHOS complexes in the (B) *soleus* and (C) EDL muscle and (D) mtDNA copy number in the *soleus* muscle of 12-week old mice, analysed by individual 2-way ANOVAs for a main effect of genotype and a main effect of sex, n = 11–13. (E) *Dnmt3a* expression in primary myotubes isolated from male WT and mD3AKO mice at differentiation day 0, 1, 3 and 5 compared to the housekeeping gene *18S*, fold-change compared to WT, n = 3, analysed by individual t-tests, corrected for multiple comparisons. (F) Oxygen consumption rate and (G) extracellular acidification rate determined using the Seahorse XF Analyser Mito Stress Test in the same myotubes at differentiation day 5. Drug treatments were oligomycin, FCCP and antimycin A + rotenone. n = 3, analysed by individual t-tests of the average of each state. ♀♂ P < 0.05 main effect of sex, ** P < 0.005. Data show the mean (bar) with individual data points or mean +SD.

## Loss of DNMT3A alters methylation and transcription of genes involved in skeletal muscle development

We found 56 genes differentially expressed between WT and mD3AKO *soleus* muscle (Fig 7A). Of those, 43 genes were significantly upregulated while 13 were significantly downregulated (Fig 7B). The most upregulated gene was *Myh8* (~11-fold increase compared to WT) while the most downregulated was *Dnmt3a*. When percent methylation of promoter regions was compared to log fold-changes (log FC) of the corresponding transcripts, comparing mD3AKO muscle to WT muscle, there was a small but significant negative correlation (R = -0.026) suggesting that transcriptional changes between WT and mD3AKO muscle are on the whole not caused by altered promoter methylation (Fig 7C). Altered gene body methylation (comparing muscle between the different genotypes) had a stronger negative correlation with RNA abundance than promoter methylation (R = 0.046, Fig 7D). Promoter and gene body methylation of differentially expressed genes after exercise displayed a similar lack of correlation to gene expression (S4 Fig). We found 8 genes which displayed both significantly altered DNA methylation (either at the promoter region or within the gene body) and altered transcript level between WT and mD3AKO *soleus* muscle, all of these had a reduction in DNA methylation and an increase in expression when comparing between the muscle of mD3AKO and WT mice (Fig 7E). Gene ontology analysis of differentially expressed genes revealed enriched terms mostly related to a downregulation of DNA methylation and DNA methylation

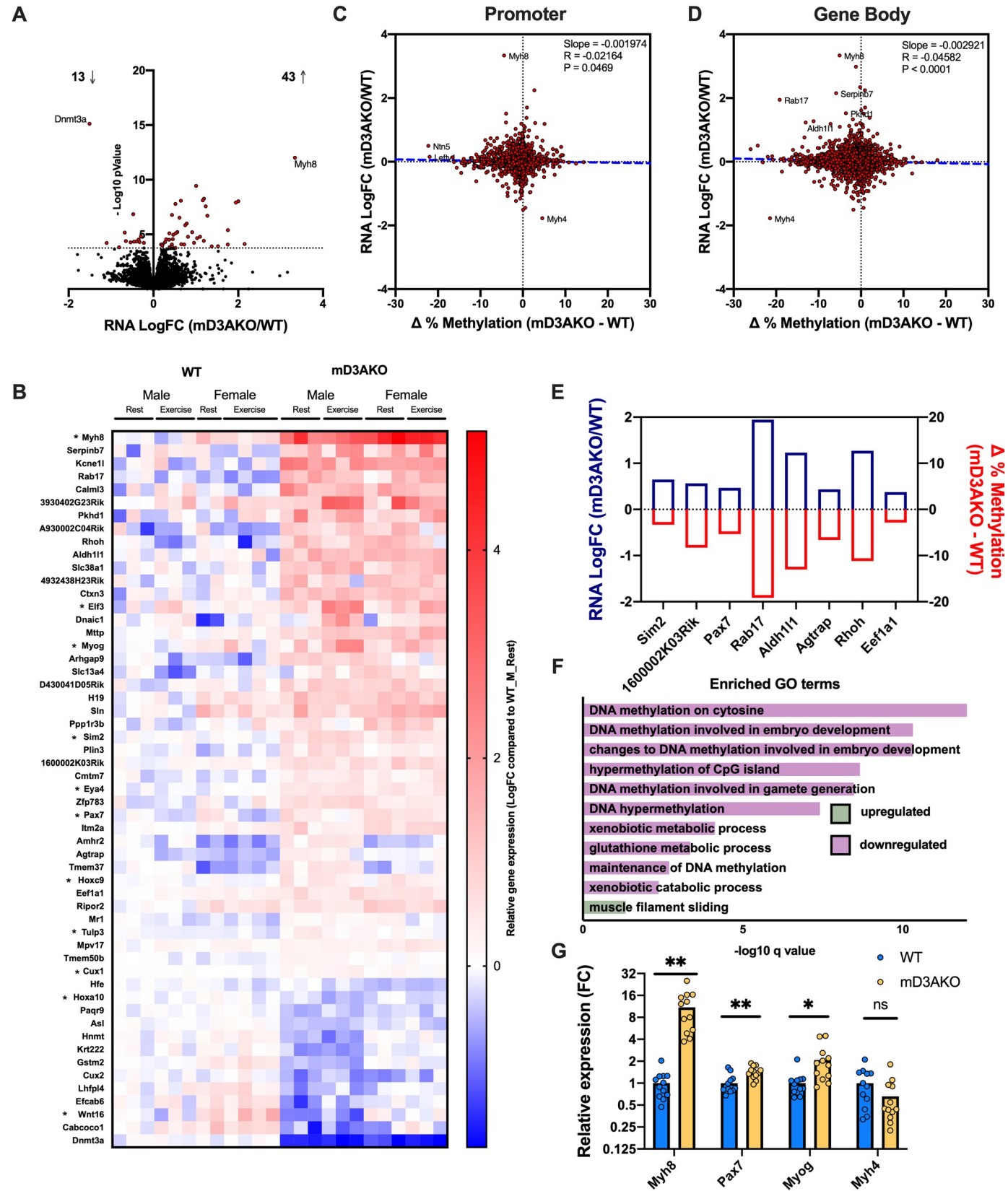

**Fig 7. Ablation of DNMT3A in skeletal muscle alters transcription of differentially methylated genes and genes involved in embryonic development.** (A) Volcano plot of all transcripts between the muscle of WT and mD3AKO mice, significant differential expression is indicated by red dots. (B) Heatmap of differentially expressed transcripts between WT and mD3AKO muscle, displayed as LogFC compared to the mean of the WT male rest group, * represent developmental genes. Change in % methylation in (C) promoter regions and (D) gene bodies vs LogFC of RNA abundance of mD3AKO muscle compared to WT muscle, showing significant inverse correlation (simple linear regression). (E) Genes that are both significantly differentially expressed and have significantly altered DNA methylation within the promoter region or gene body between WT and mD3AKO muscle. (F) Significant GO-BP (gene ontology—biological processes) terms from the differentially expressed transcripts. (G) mRNA abundance of significantly differentially expressed genes involved in muscle development from RNA-seq data, expressed as fold over WT. From *soleus* muscle of 12-week old male and female WT and mD3AKO mice, n = 12, * Padj < 0.05, *** Padj < 0.0005. Data show the mean (bar) with individual data points.

relating to embryonic development (Fig 7F). Additionally, genes related to "metabolism of xenobiotics and glutathione" were significantly downregulated while genes related to the term "muscle filament sliding" were significantly upregulated. Several genes involved in muscle cell development including *Myh8*, a perinatally expressed myosin heavy chain isoform and *Pax7* and *Myog*, two transcription factors crucial for normal muscle cell development, were significantly upregulated in the skeletal muscle of mD3AKO mice (Fig 7G). Genes involved in the "development" gene ontology term have a star next to their gene symbol (Fig 7B).

In order to validate some of the differentially methylated/expressed genes found in *soleus* muscle (Fig 7E) in other muscle types, we performed targeted bisulfite pyrosequencing and RT-qPCR on the EDL (white, mainly type II fibers) and *quadriceps* (mixed, type I and II fibers) muscles from a separate cohort of mice. We included *H19* in this analysis, due to presence of a differentially methylated CpG site in the gene body and excluded *Rhoh* and *Rab17* due to low expression levels (CPM < 1), *Sim2* due to technical issues with pyrosequencing as well as the uncharacterized gene, *1600002K03Rik*. Similarly to the RRBS data in *soleus* muscle (Fig 8A), all the regions studied (*H19*, *Aldh1l1*, *Agtrap*, *Pax7*, and *Eef1a1*) showed a clear and significant reduction in methylation when comparing mD3AKO to WT in both the EDL and *quadriceps* muscles in all of the CpG sites studied, apart from the first CpG measured in the *H19* cluster (Fig 8B). However, when examining levels of the corresponding transcripts, only 3 of the 5 targets (*H19*, *Aldh1l1* and *Agtrap*) displayed a similar increase in expression in the EDL and *quadriceps* muscles (Fig 8D) compared to the *soleus* muscle (Fig 8C) when comparing mD3AKO to WT mice. Together, these data suggest that DNMT3A targets regions in multiple muscle types that are associated with muscle type-dependent increases in gene transcription.

## Discussion

There is substantial evidence that exercise and diet can alter DNA methylation in skeletal muscle tissue. However, it is currently unclear both by which mechanisms DNA methylation is altered and if these changes influence skeletal muscle physiology. In order to investigate the role of *de novo* DNA methylation in these processes, we utilized a mouse model with DNMT3A ablated specifically in mature muscle (mD3AKO mice). Due to the major role of DNMT3A in embryonic and early-life development (most whole-body DNMT3A-null mice die before 4-weeks of age [20]), we wanted to use a model in which DNMT3A was knocked out only in fully differentiated skeletal muscle fibers (Fig 6E) in order to remove the possibility that exercise or metabolic phenotypes were driven by changes in muscle differentiation. DNMT3A protein levels were also reduced in the heart however, as few differences were found in whole-body energy metabolism or the response to exercise, we did not further examine the impact of DNMT3A ablation in heart.

Knockout of DNMT3A substantially altered the DNA methylome of skeletal muscle, evidenced by robust separation of genotypes after principal component analysis. This separation was far greater than the separation following exercise, which suggests that ablation of DNMT3A has a greater impact than exercise on DNA methylation in skeletal muscle.

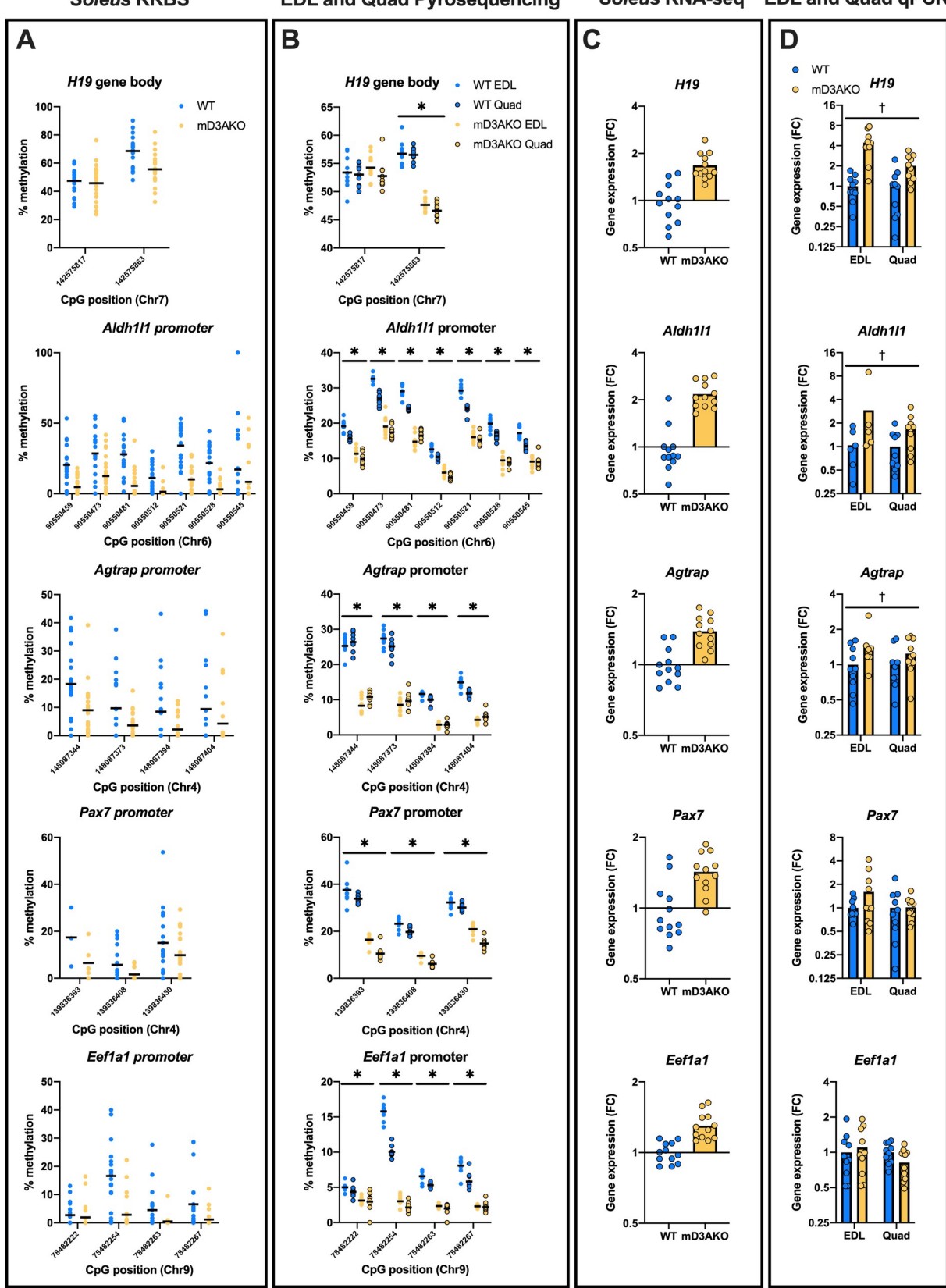

**Fig 8. Hypomethylation of specific regions in mD3AKO muscle is consistent over multiple skeletal muscle types.** Analysis of gene body/ promoter methylation and transcription of putative DNMT3A target genes, *H19*, *Aldh1l1*, *Agtrap*, *Pax7* and *Eef1a1*. (A) CpG methylation of significantly differentially methylated gene body/promoter regions from *soleus* RRBS data, FDR 5%, n = 3–24 (some CpG sites were only detected in some samples). (B) CpG methylation of the same regions from targeted bisulfite pyrosequencing in the EDL and *quadriceps* muscles of a separate cohort of male WT and mD3AKO mice, n = 10, analysed by individual t-tests between WT and mD3AKO for each muscle, corrected for multiple comparisons (total number of CpG sites analysed). (C) Significantly differentially expressed transcripts from the corresponding genes from *soleus* RNA-seq data, FC compared to WT, n = 12, FDR 5%. (D) Gene expression of the same genes by qPCR in the EDL and *quadriceps* muscles compared to the housekeeping gene *18S* of a separate cohort of male WT and mD3AKO mice, FC compared to WT, n = 5–10 (due to low expression of some genes, samples with CT values over 40 were excluded), analysed by 2-way ANOVA for a main effect of genotype. * AdjP < 0.0001 for each CpG site (both muscles). † P < 0.05 main effect of genotype. Data show the mean with individual data points.

However, global DNA methylation measured as a percentage of quantified CpG sites was reduced by an average of only ~1% (WT mice ~25%, mD3AKO mice ~24%). This suggests that a large proportion of DNA methylation in skeletal muscle is static after development and this methylation is likely maintained by DNMT1 rather than the *de novo* methyltransferases. Interestingly, this loss of DNA methylation occurs in all genomic contexts studied which suggests that *de novo* methylation catalysed by DNMT3A is not specific to promoters or CGIs. Potentially, the minor global hypomethylation of DNA caused by DNMT3A ablation may indicate a loss of the maintenance function of DNMT3A [21]. Supporting this idea is the differential methylation and transcription of the imprinted gene *H19*, which is normally maintained in a semi-methylated state (paternal allele is methylated). Despite this, if we plot change in promoter DNA methylation, there is a clear bias for a loss of methylation if we compare mD3AKO to WT muscle (Fig 1F).

We found no difference in either voluntary or forced exercise capacity between WT and mD3AKO mice. Similarly, the transcriptional response to acute exercise in skeletal muscle was the same between genotypes suggesting that *de novo* methylation by DNMT3A does not mediate exercise-responsive transcription. Comparably, we found that acute exercise causes a much more robust effect on the transcriptome than on the methylome. This would suggest that much of the transcriptomic response to acute exercise may not be driven by *de novo* CpG methylation/demethylation. A similarly small overlap between significantly altered promoter methylation and corresponding transcripts after exercise was found in a previous investigation performed by our lab in human skeletal muscle [7]. As acute exercise is associated with demethylation of exercise-responsive genes [3], ablation of enzymes participating in DNA demethylation such as the TET enzymes, instead of the *de novo* DNA methyltransferases, may have caused more effects on the transcriptome after exercise. Yet, it was speculated that *de novo* DNA methyltransferase activity was needed to remethylate exercise-demethylated gene promoters and shut down transcription [3]. If such a mechanism is at play, our data showing that transcription after exercise is not altered in skeletal muscle lacking DNMT3A suggest that DNMT3A does not participate in *de novo* methylation of exercise-demethylated genes.

As skeletal muscle is responsible for a large proportion of postprandial glucose disposal [22] we examined glucose metabolism in mD3AKO mice. Similar to a previous report, ablation of DNMT3A in the muscle of chow-fed mice did not affect glucose tolerance or fasting blood glucose levels [23]. Additionally, there was no difference in glucose tolerance between genotypes in male mice fed a HFD for a period of 12 weeks despite an expected significant increase in body weight and fat mass and a worsening of glucose tolerance in HFD-fed mice compared to chow-fed mice. This suggests that DNMT3A does not impact on the adaptation of muscle to a high-fat diet. This was not the case in adipose tissue where ablation of DNMT3A has been reported to improve glucose tolerance in mice fed a HFD, but not in chow-fed mice [17]. Unlike the adipose tissue which displays a clear upregulation of *Dnmt3a* expression in obese

mice or mice fed a HFD [17,24], we found little difference in *Dnmt3a* expression in skeletal muscle between age matched chow and 16–18 week HFD-fed mice. This is opposed to a study reporting a ~2-fold increase in *Dnmt3a* expression in the skeletal muscle of 3-month HFD-fed mice compared to chow-fed controls [17].

There are several reports that suggest DNMT3A may have a role in influencing mitochondrial DNA (mtDNA) methylation [19,25,26], potentially impacting mitochondrial function. However, the presence of cytosine methylation in mtDNA has been questioned and may be largely overestimated due to incomplete bisulfite conversion of coiled mtDNA [27]. The other *de novo* DNA methyltransferase isoform, DNMT3B, plays a role in the regulation of mitochondrial density in skeletal muscle cells by fatty acid-induced methylation of the *PPPARC1A* promoter [8]. In the current study, no differences were found in skeletal muscle mitochondrial content, mitochondrial function in isolated muscle cells or energy expenditure at a whole-body level between WT and mD3AKO mice suggesting that DNMT3A is not important for the regulation of mitochondrial function or density in mature skeletal muscle.

One advantage of the current study is that DNA methylation and mRNA abundance were determined from the same *soleus* muscle and therefore the relationship between altered DNA methylation and transcription can be examined. DNA methylation, both at promoter regions and in gene bodies, displayed a slight inverse correlation with transcript levels of the corresponding genes, with a stronger correlation with gene bodies than promoter regions. However, after multiple correction, only 8 genes displayed both a significant reduction in methylation (in the promoter region or gene body) and a corresponding increase in gene expression (Fig 7E). A similar lack of correlation between changed promoter methylation and gene transcription was observed in genes that were differentially expressed during exercise. This is comparable to reports that describe a low overlap between differentially methylated promoter regions and differentially expressed transcripts after exercise training [7,28] or exposure to a short term HFD in human skeletal muscle [11]. It is possible that differential methylation of genomic enhancer regions upstream or downstream of promoter regions may have a larger role in regulation of transcription which were not captured with this analysis, due to the bias of RRBS towards CpG islands. Genes that did change with genotype were part of gene ontology sets related to DNA methylation, glutathione metabolism and muscle filament sliding. Altered glutathione metabolism may be a consequence of reduced DNA methylation as the substrate for cytosine methylation, s-adenosyl methionine shares the same amino acid precursors as glutathione: methionine and homocysteine [29]. Together, this dataset greatly expands the known DNMT3A targets in skeletal muscle [23,30] (evidenced by reduced promoter/gene-body methylation and a corresponding increase in the transcript level) and indicates possible enhancer interactions governing the other differentially expressed genes. Additionally, 5 putative DNMT3A targets (*H19*, *Aldh1l1*, *Agtrap*, *Pax7* and *Eef1a1*) were validated in multiple skeletal muscles (*soleus*, EDL and *quadriceps*) displaying significant promoter/gene body hypomethylation in mD3AKO muscle. In 3 of the targets (*H19*, *Aldh1l1* and *Agtrap*), this was accompanied by an increase in the transcript level of the corresponding gene in all of the muscles studied.

There have now been several investigations utilizing muscle-specific DNMT3A knockout mice. When DNMT3A is knocked out employing Cre recombinase driven by a muscle satellite cell marker (Pax3), mice are smaller due to substantially reduced skeletal muscle mass and muscle regeneration is compromised [30]. When the Cre recombinase is driven by a muscle marker that is intermediate in muscle cell development (alpha-actin), mice have normal body and skeletal muscle mass however muscle regeneration is still compromised [23]. In the current study, Cre recombinase was driven by the muscle creatine kinase promoter which is expressed late in skeletal muscle cell development [16], and no effect of DNMT3A knockout was seen on body weight, skeletal muscle size or morphology, presumably as DNMT3A is still

present in satellite cells. However, loss of DNMT3A in skeletal muscle altered the transcription of genes involved in embryonic development and morphogenesis (*Sim2*, *Eya4*, *Hoxc9*, *Cux1*, *Hoxa10*, *Wnt16*, *Elf3*) as well as upregulating genes involved in muscle cell development such as *Myh8*, *Pax7* and *Myog*. Overexpression of the myogenic transcription factors *Myog* [31] and *Pax7* [32] has been reported to inhibit skeletal muscle growth and differentiation. This suggests that earlier deletion of DNMT3A in the process of skeletal muscle differentiation may have a more substantial effect on muscle function.

In conclusion, ablation of DNMT3A in differentiated skeletal muscle does not affect skeletal muscle size, exercise capacity, glucose metabolism or mitochondrial content or function despite having a substantial effect on muscle DNA methylation. Additionally, this investigation revealed that changes in DNA methylation and gene transcription in skeletal muscle in response to acute exercise are not driven by DNMT3A and identified multiple novel DNMT3A targets over different skeletal muscle types validated by RRBS, bisulfite pyrosequencing and gene expression. Together with previous investigations highlighted above, our data suggest that the major role of DNMT3A in skeletal muscle is likely to be in the development and differentiation of muscle tissue either during embryonic development or muscle regeneration from satellite cells, rather than a role in mature skeletal muscle function.

## Materials and methods

### Ethics statement

Experiments involving mice were approved by the Danish Animal Experiments Expectorate and performed according to local committee guidelines (license numbers 2011/561-92 and 2019-15-0201-01638).

### Animals

To excise *Dnmt3a*, LoxP sites flanked exon 19 of *Dnmt3a*, which codes for a PC motif in the catalytic domain. Cre-mediated recombination would thereby lead to a frameshift for all *Dnmt3a* variants as previously described [33]. Muscle specific DNMT3A knockout mice were generated by crossing mice heterozygous for muscle creatine kinase promoter-driven Cre recombinase (*mCK-Cre*$^{-/+}$) with mice homozygous for floxed *Dnmt3a* (*Dnmt3a*$^{flox/flox}$, gift from Prof En Li from the Cardiovascular Research Center of Massachusetts General Hospital, USA) on a C57BL/6 background. This cross produced the experimental animals, genotyped as either WT (*Dnmt3a*$^{flox/flox}$) or mCK-DNMT3A (*Dnmt3a*$^{flox/flox}$ *mCK-Cre* Tg). Mice were group-housed with littermates of the same sex (2–7 in a cage), maintained at 22˚C±1˚C, and were fed either chow diet (Altromin diet #1310) or a high-fat diet (HFD, 45% calories as fat, Research Diets D12451). Mice had *ad libitum* access to food and water and were kept at a 12:12h light:dark cycle. Apart from the mice in the acute exercise protocol detailed below, mice were killed between 36–40 weeks of age at 14:00 after a 6 hour fast by intraperitoneal pentobarbital overdose (100 mg/kg of body weight) and cervical dislocation, blood was taken by cardiac puncture (prior to cervical dislocation) and muscle tissues were rapidly dissected and snap-frozen.

### Acute exercise protocol for transcriptomics and methylation analysis

12-week-old male and female mice were used in these experiments. Mice were familiarized to a treadmill (Exer3/6, Columbus Instruments) for three consecutive days with 10 min of treadmill running with 0% incline at 10 m/min. On the day of acute exercise mice performed 30 min of treadmill running with 5% incline at 16m/min. Mice were sacrificed by cervical

dislocation at 10, 30 or 60 min immediately after completion of treadmill running. A separate group of mice were sacrificed at the same time after 30 minutes on a sham treadmill arena dedicated as "rest". *Soleus* muscles were dissected and snap frozen in liquid nitrogen for later analysis. RRBS was performed on the *soleus* muscle from all timepoints while RNA-seq was performed on the "rest" and 60 minutes after exercise timepoints.

### Exercise tolerance test and voluntary wheel running

32–36 week-old chow-fed mice were familiarized to the treadmill for three consecutive days with 10 min of treadmill running with 0% incline at 10 m/min finishing 2 days before the exercise tolerance test. On the day of the exercise tolerance test, at 14:00, mice underwent a graded treadmill test, at a 5% incline, in which they were warmed up with a speed of 10 m/min for 10 minutes and then every 4 minutes speed was increased by 2 m/min to a maximum of 30 m/min. Mice that lagged at the charged coil for a period of 10 seconds were taken off the treadmill and the running time and distance were recorded. Blood lactate was determined by hand-held lactate meter (Lactate Pro 2, Axonlab) from tail blood, prior to and immediately following treadmill running. A separate cohort of 32–36 week-old chow-fed mice were used for voluntary wheel running experiments. Mice were singly housed and acclimatized to wheel running cages for 1 week. After acclimatization, mice were kept in running wheel cages for an additional 2 weeks and running distance and speed were monitored using bicycle computers.

### Assessment of body composition, indirect calorimetry and oral glucose tolerance test

Body composition and glucose tolerance were determined in mice at 30 weeks of age (12 weeks of high-fat feeding for HFD mice). Lean and fat mass were measured using the EchoMRI-500 (EchoMRI LLC, Houston, USA) according to the manufacturers' instructions, excluding body water. For the oral glucose tolerance test, mice were fasted for 6 hours (food removed at 8:00, oGTT started at 14:00) and glucose was administered by gavage with a laryngeal cannula corresponding to 2.5g of glucose/kg of lean mass. Blood glucose levels were monitored from the tail tip using a hand-held glucometer (Contour XT, Bayer) before, and for 90 minutes following glucose administration. Indirect calorimetry was performed in 31–32 week old mice. Mice were acclimatized to the Home Cage System Phenomaster (TSE systems) cages (without running wheels) for 1 week prior to recording measurements. Gas exchange was monitored every 20 minutes for a period of 5 days and was averaged to hourly intervals for a 24-hour period.

### Plasma analysis

Plasma insulin was analysed by ELISA (Ultra Sensitive Mouse Insulin ELISA kit, Crystal Chem, #90080). Plasma NEFA was analysed by NEFA HR(2) kit (Wako).

### Immunoblotting

Tissue was homogenized in RIPA buffer containing protease (SIGMAFAST Protease Inhibitor Cocktail Tablet, Sigma) and phosphatase inhibitors (10 mM NaF, 1 mM $Na_3VO_4$) by steel bead disruption (3 x 90 sec at 30 Hz using the TissueLyser II, Qiagen). Protein amounts were normalized after determination of protein concentration using BCA protein assay kit (Pierce) in a standard laemmli buffer and subjected to SDS-PAGE, transferred to a PVDF membrane and blotted for DNMT3A (Abcam, #ab188470) or OXPHOS rodent cocktail (Abcam, #ab110413). Protein loading was determined using Bio-Rad stain-free technology.

## RNA and DNA isolation

Muscle tissue was homogenized in 350–600 µl of RLT plus buffer (Qiagen) with β-mercaptoethanol using steel beads and subjected to 3 x 90 sec at 30 Hz of disruption using the TissueLyser II (Qiagen). RNA and DNA were simultaneously isolated using the RNA/DNA/miRNA AllPrep kit (Qiagen) as per instructions. Primary muscle cells were lysed directly in RLT buffer and RNA extracted using the RNeasy kit (Qiagen). RNA was eluted in 45 µl of RNase free water and DNA in 60 µl of elution buffer (Qiagen).

## RNA sequencing

RNA was checked for quality using the Bioanalyser instrument (Agilent Technologies) and subject to the Illumina TruSeq Stranded Total RNA with Ribo-Zero Gold protocol (Illumina) and performed as described. Libraries were subjected to 75-bp paired-end sequencing on NextSeq 550 (Illumina). Approximately 32 million reads/sample were assigned to genes with 17389 genes surviving the expression threshold.

## Genome-wide DNA methylation analysis

Reduced Representation Bisulfite Sequencing (RRBS) was performed as described [7]. Briefly, DNA samples were incubated with MspI enzyme (NEB) in order to fragment DNA at CCGG positions to enrich for CpG regions. Bisulfite conversion was performed using the EZ DNA methylation Kit (Zymo Research) according to the manufacturer's instructions. DNA was then PCR amplified and ligated to TruSeq (Illumina) sequencing adapters. Libraries were subjected to 75-bp single-end sequencing on a NextSeq 550 (Illumina). On average, 18.3 million reads/sample were aligned and 205304 CpGs had sufficient coverage in every sample.

## Bioinformatic analysis

RNA sequencing reads were processed as previously described [34], with the difference that the gene model used was mm10 GENCODE Comprehensive gene annotations version 15 primary assembly (https://www.gencodegenes.org/mouse/release_M15.html). RRBS reads were preprocessed with Trim Galore v. 0.4.0 (https://github.com/FelixKrueger/TrimGalore), a wrapper for cutadapt v. 1.9.1 [35], with the—rrbs flag set and aligned to the mm10 genome assembly using Bismark v. 0.18.1 [36], with default parameters and finally CpG methylation calls were summarized using the bismark2bedgraph script which is part of the Bismark software suite. Samples with less than 10 million aligned reads were observed to have different global levels of methylation and were excluded from further analysis. For each CpG, observations from both strands were added together to form an overall methylation level per CpG, and only CpGs covered by eight or more reads in every sample was used for testing for differential methylation. Chromosomes M, X and Y were excluded, the former due to the lack of methylation on chromosome M and the latter two since the two sexes were not equally represented in all groups and the sex differences were not of interest. Testing for differential methylation was performed using the procedure described by Chen *et al*. version 2 [37], briefly the ratio between methylated and unmethylated counts is modelled by a negative binomial linear model as implemented in the edgeR framework [38]. The linear model consisted of the main effects of genotype, time post exercise and an interaction between the two. In addition to differential methylation analysis at the single CpG level, differential methylation was also tested using counts aggregated within promoters, defined as 3000 bp up stream and 1000 bp downstream of TSS, and gene bodies both defined by the same gene model used for RNA-seq analysis. Finally, CpGs were also aggregated and tested by CpG clusters where each cluster was

defined by scanning along the genome and grouping CpGs no more than 100bp apart. The fraction of methylation was calculated for each CpG included in the analysis and used both to annotate the differential methylation results and for performing a principal component analysis of the sample methylation levels. Gene ontology of differentially methylated clusters was performed using the Genomic Regions Enrichment of Annotations Tool (GREAT) [39] using the standard settings. *Soleus* mtDNA copy number was calculated from RRBS data by determining the ratio of reads that aligned to chromosome M compared to total aligned reads, as previously described [40].

## Targeted bisulfite pyrosequencing

Bisulfite-pyrosequencing was performed on EDL and *quadriceps* samples from 36–40 week-old male WT and mD3AKO mice. 1 µg of genomic DNA was bisulfite-converted using the EZ DNA Methylation-Lightning kit (Zymo Research) and 30 ng of bisulfite-converted DNA per assay was amplified with the Qiagen PyroMark PCR kit following the manufacturer's instructions. Amplicons were sequenced on a PyroMark Q48 Autoprep instrument (Qiagen). Pyrosequencing primers were designed using the Qiagen PyroMark Assay Design Software 2.0 and are displayed in S1 Table. Targets were chosen from differentially methylated clusters from the *soleus* RRBS data. Data were analysed with the provided PyroMark Q48 Autoprep software.

## cDNA synthesis and qPCR

cDNA was synthesized using the Bio-Rad iScript cDNA synthesis kit from 1 µg of RNA. qPCR was performed using conventional Sybr Green chemistry utilizing primers displayed in S1 Table. All primers were used at a final concentration of 200 nM. Relative quantification was determined by comparing samples to a standard curve of pooled cDNA and normalized to the housekeeping gene *18S*.

## Primary muscle cells

Muscle satellite cells were isolated from the complete hindlimbs of 30-week-old male mice (3 WT and 3 mD3AKO) as described [41]. Satellite cells were seeded at 20,000 cells/well (96-well plate format) on Seahorse XFe96 Cell Culture Microplates coated with 1% Matrigel basement membrane (Corning) and differentiated into myotubes in high glucose DMEM with 2% horse serum for 5 days. Real-time measurements of oxygen consumption rate (OCR) and extracellular acidification rate (ECAR) were performed using a Seahorse XFe96 Extracellular Flux Analyzer (Agilent Technologies) utilising the Seahorse XF Cell Mito Stress Test kit (Agilent Technologies). OCR and ECAR were measured under basal conditions and after injection of final concentrations of 1 µM oligomycin, 2.3 µM FCCP, or 2.55 µM antimycin A combined with 1 µM rotenone. The measured values were normalized to protein levels by lysing the cells in 50 mM NaOH (70°C for 1 hour) and performing a BCA protein assay. 14–32 technical replicates (wells) were measured for each of the 3 primary muscle cell lines per genotype and an average was taken. The same satellite cells were grown and differentiated in parallel in 12-well plates for RNA extraction.

## Analysis of myofiber size

*Plantaris* muscles from 12-week-old mice were frozen in isopentane and embedded in paraffin blocks. 2 µm muscle sections were stained using Picrosirius red (Direct Red 80, Sigma) for 15 minutes and dehydrated in 99% ethanol. Slides were imaged on a Zeiss Primovert light microscope with the objective lens at 20x magnification. Myofiber cross sectional area (CSA) was

calculated using ImageJ software (National Institutes of Health, Bethesda, MD) from at least 4 fields of view per slide and the average CSA per mouse was calculated.

## Data availability

RNA-seq and RRBS data are archived for public access at the Gene Expression Omnibus (https://www.ncbi.nlm.nih.gov/geo/query/acc.cgi?acc=GSE152349) under accession number GSE152349. A summarised spreadsheet of the RNA-seq data is included in the supplemental information (S1 Spreadsheet). All of the numerical data underlying the figures is available in the supplemental information (S2 Spreadsheet).

## Statistical analysis

Data are expressed as means, error bars depict standard deviation. For mouse experiments utilising both female and male mice, results were analysed by 2-way ANOVA for a main effect of genotype and a main effect of sex. When exercise was included as a factor (Fig 2), both sexes were analysed individually by 2-way ANOVA for a main effect of genotype and a main effect of exercise. When only genotype was tested, results were analysed by a student's t-test. When comparing between different muscle types, results were analysed by 2-way ANOVA for a main effect of genotype and a main effect of muscle. N number represents individual animals or cell lines derived from individual animals. Statistical analysis was performed in GraphPad Prism software (Prism 8). A p-value of less than 0.05 was considered statistically significant. Differential expression and differential methylation data were analysed for a main effect of genotype, a main effect of exercise and a genotype/exercise interaction. As there were no significant exercise/genotype interactions in either the RNA-seq or RRBS datasets, exercised and non-exercised mice were grouped together for the analysis performed in Fig 7. A false discovery rate (FDR) of less than 5% was considered significant.

## Supporting information

**S1 Fig. Skeletal muscle expression of *Dnmt3a* and gene ontology of differentially methylated regions between WT and mD3AKO muscle.** (A) *Dnmt3a* expression in the EDL and *quadriceps* muscles of 36–40-week-old male WT and mD3AKO mice compared to the housekeeping gene *18S*, fold-change compared to WT, n = 10. Analysed by individual t-tests. Significant GO-BP (gene ontology—biological processes) terms from hypomethylated (B) clusters and (C) promoter regions when comparing mD3AKO to WT *soleus* muscle. There were no significant terms for hypermethylated regions. FDR 5%. *** P < 0.0005. Data are the mean (bar) with individual data points.
(TIF)

**S2 Fig. Modest changes in DNA methylation in muscle after exercise.** (A) Number of significantly differentially methylated regions in the *soleus* muscle when comparing genotype, time after exercise and a genotype/exercise interaction. (B) Percentage of total methylated cytosine residues quantified by RRBS in the *soleus* muscle of non-exercised mice (rest) or taken 10, 30 or 60 minutes after 30 minutes of treadmill running, combined male and female. Analysed by 2-way ANOVA, n = 6–7. Data are the mean (bar) with individual data points.
(TIF)

**S3 Fig. No difference in skeletal muscle size or morphology between mD3AKO and WT mice.** (A) *Soleus* and *tibialis* muscle weights of 36–40 week-old male and female WT and mD3AKO mice, n = 15–21. (B) Representative sirius red staining of the *plantaris* muscle of WT and mD3AKO mice from 12-week-old mice and (C) myofiber cross-sectional area,

n = 7–9. Analysed by 2-way ANOVA for a main effect of genotype and a main effect of sex. (D) Gene expression of markers of slow and fast twitch fibers from the *soleus* muscle of 12-week old mice (male and female combined) normalised to WT, n = 12, analysed by individual t-tests. ♀♂ P < 0.05 main effect of sex. Data are box plots or the mean (bar) with individual data points.
(TIF)

**S4 Fig. Correlations between the expression of differentially regulated transcripts by exercise and the DNA methylation of corresponding regions.** Change in % methylation in promoter regions and gene bodies at (A, D) 10, (B, E) 30 and (C, F) 60 minutes after 30 minutes of treadmill running compared to rested muscle vs LogFC of RNA abundance in muscle 60 minutes after 30 minutes of treadmill running compared to rested muscle. A significant correlation is demonstrated by a blue line.
(TIF)

**S1 Spreadsheet. Summarized RNA-seq data of the *soleus* muscle.** Displaying main effect of genotype, main effect of exercise and a genotype/exercise interaction.
(XLSX)

**S2 Spreadsheet. Raw numerical data from all figures.** Raw data is displayed in an Excel spreadsheet with one sheet for each figure and supplemental figure.
(XLSX)

**S1 Table. Table of qPCR and pyrosequencing primer sequences used in this study.**
(DOCX)

## Acknowledgments

The authors would like to thank the staff of the Department of Experimental Medicine, University of Copenhagen for assistance with animal care. We acknowledge the Single-Cell Omics platform at the Center for Basic Metabolic Research (CBMR) for technical and computational expertise and support.

## Author Contributions

**Conceptualization:** Lewin Small, Rhianna C. Laker, Romain Barrès.

**Data curation:** Lewin Small, Lars R. Ingerslev.

**Formal analysis:** Lewin Small, Lars R. Ingerslev.

**Funding acquisition:** Lewin Small, Romain Barrès.

**Investigation:** Lewin Small, Eleonora Manitta, Rhianna C. Laker, Ann N. Hansen, Brendan Deeney.

**Methodology:** Lewin Small, Lars R. Ingerslev, Eleonora Manitta, Rhianna C. Laker, Romain Barrès.

**Project administration:** Romain Barrès.

**Resources:** Alain Carrié, Philippe Couvert, Romain Barrès.

**Software:** Lars R. Ingerslev.

**Supervision:** Romain Barrès.

**Validation:** Lewin Small, Lars R. Ingerslev.

**Visualization:** Lewin Small.

**Writing – original draft:** Lewin Small, Romain Barrès.

**Writing – review & editing:** Lewin Small, Lars R. Ingerslev, Eleonora Manitta, Rhianna C. Laker, Ann N. Hansen, Philippe Couvert, Romain Barrès.

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
