## [Decision Letter · Decision Letter 0]

2 Oct 2020

Dear Dr Barres,

Thank you very much for submitting your Research Article entitled 'Ablation of DNA-methyltransferase 3A in skeletal muscle does not affect energy metabolism or exercise capacity' to PLOS Genetics. Your manuscript was fully evaluated at the editorial level and by three independent peer reviewers. The reviewers appreciated the attention to an important problem, but raised some substantial concerns about the current manuscript. Based on the reviews, we will not be able to accept this version of the manuscript, but we would be willing to review again a much-revised version. We cannot, of course, promise publication at that time.

Should you decide to revise the manuscript for further consideration here, your revisions should address the specific points made by each reviewer. In particular, the comments raised by reviewers 1 and 3 need your further attention. We will also require a detailed list of your responses to the review comments and a description of the changes you have made in the manuscript.

If you decide to revise the manuscript for further consideration at PLOS Genetics, please aim to resubmit within the next 60 days, unless it will take extra time to address the concerns of the reviewers, in which case we would appreciate an expected resubmission date by email to plosgenetics@plos.org.

[LINK]

We are sorry that we cannot be more positive about your manuscript at this stage. Please do not hesitate to contact us if you have any concerns or questions.

Yours sincerely,

Yvonne Böttcher, Ph.D.

Guest Editor

PLOS Genetics

Wendy Bickmore

Section Editor: Epigenetics

PLOS Genetics

Reviewer's Responses to Questions

**Comments to the Authors:**

Reviewer #1: In this study, the authors systematically examine gene expression and DNA methylation with or without exercise in Ck-induced Dnmt3a cKO mice to see the potential function of DNA methylation status in muscle in exercise.

Figure 7 represents the phenotypical difference between WT and KO mice, however, samples of non-exercised ground and exercised group were mixed together for the analysis. The expression and methylation status should be examined separately between WT versus KO at rest and WT versus KO during exercise.

DNA methylation status in skeletal muscle in aging is characterized by overall hypomethylation and partial promoter hypomethylation.

Considering that the presented list includes the imprint H19 and Sim2 among the hypomethylated genes, this methylation change may be due to a deletion of the maintenance function of Dnmt3a. This should be disucssed. (The function of Dnmt3a in maintaining DNA methylation, https://pubmed.ncbi.nlm.nih.gov/19789556/)

Reviewer #2: The authors investigated the function of de novo DNA methylation in fully differentiated skeletal muscle with muscle specific DNA methyltransferase 3A (DNMT3A) knockout mice and investigated the impact of DNMT3A ablation on skeletal muscle DNA methylation, exercise capacity and energy metabolism. The study is very well written and rigorously performed. I very much enjoyed reading the paper and the concept is interesting. The results suggest that use of this model would not unravel the effect of exercise on DNA methylation, hence the focus should perhaps be on human skeletal muscle (single fibre?) models to better understand the effect of exercise on the methylome.

So, overall by reporting these negative results, this is an important report that would add to the current literature.

My minor questions and comments:

1. How is the RNA-seq add to this report if the DNA methylation model did not show direction? Have the transcriptome data was performed at baseline alone? When you stated ‘We

found 8 genes which displayed both significantly altered DNA methylation’ is this after exercise?

2. I would add some discussion on the transcriptome at the discussion section.

Regards,

Nir Eynon

Reviewer #3: Summary

The study higlights the impact of skeletal muscle-specific Dnmt3a deficiency on soleus methylation and gene expression, exercise capacity, energy metabolism and whole body metabolism.

The Key findings include:

1. Dnmt3a knock-out soleus muscle display reduced DNA methylation.

2. Muscle-specific Dnmt3a deficiency did not affect exercise capacity or the transcriptional response to acute exercise.

3. mD3AKO mice have normal body composition, whole-body energy expenditure, substrate utilization and glucose tolerance.

4. mD3AKO knockout shows no changes in mitochondrial function but alters transcription of gene(s) involved in skeletal muscle development.

The authors quite nicely performed both methylation and gene expression profiling studies and physiological studies to determine the role of skeletal muscle DNMT3A in exercise and whole-body metabolism. Individual data set is high quality and nicely done, but there are important questions in understanding a whole picture by putting the data together as addressed below.

Major points

1. It is surprising to see that there is an only small negative correlation between differentially methylated genes and differentially expressed genes by Dnmt3a deficiency. What is the correlation of the two by exercise? A poor correlation between the two suggests that methylation and transcriptional changes are not indirect. It might have something to do with the fact that RRBS is biased on the CpG islands. Thus, methylation profile of top Dnmt3a target genes should be examined using bisulfite analysis in soleus and other muscle depot(s).

2. Authors should perform some of key analogous studies using other muscles other than soleus muscle. For example, do the authors see the same pattern of regulation in DNA methylation and gene expression of the key gene targets in white or mixed muscles?

3. More thorough model validation is needed for better data interpretation: Authors stated that they used mCK-Cre to avoid developmental effect, as it is expressed quite late during skeletal muscle differentiation and Dnmt3a expression likely comes from satellite cells and non-muscle cells.”. This statement should be revisited as previous literature (doi:10.1016/s1097-2765(00)80155-0; doi:10.1073/pnas.97.7.3467) show that mCK is active embryonically, which could account for the fact that some of developmental genes were altered. Can authors provide evdience that mCK is only active in fully differentiated muscle fiber? mCK is also expressed in heart. Was there any abnormality there? Please show the mRNA and protein level of Dnmt3a KD efficiency in soleus muscle and heart in Fig 1. To claim that they avoided developmental issue, an inducible Cre should be employed.

4. There is some inconsistency in choosing muscle depots between experiments. In Fig 6, authors use EDL muscle type to study OXPHOS. But this is a glycolytic white muscle. Since they focused on soleus muscle, the same set of experiments need to be repeated with soleus. In figure 5, authors quadriceps muscles were only looked at. What is the Dnmt3a expression in soleus and EDL?

Minor points

1. The gene list of gene ontology data from differentially methylated regions were not provided in Fig. 1.

2. Can authors provide gene ontology analysis of differentially methylated regions by breaking down to hypomethylated and hypermethylated regions?

3. In Fig. 4, the authors suggest that male mice have a higher EE compared to females. However, in the treadmill running test, the WT male mice ran for much less distance compared to WT female, which seems counterintuitive. Can the authors provide explanation for that?

4. To perform the seahorse assays, authors used muscle satellite cells, which are considered muscle stem or progenitor cells. This should be done with differentiated muscle cells either in vitro or ex vivo.

**Have all data underlying the figures and results presented in the manuscript been provided?**

Reviewer #1: Yes

Reviewer #2: Yes

Reviewer #3: Yes

PLOS authors have the option to publish the peer review history of their article (what does this mean?). If published, this will include your full peer review and any attached files.

Reviewer #1: No

Reviewer #2: No

Reviewer #3: No

---

## [Decision Letter · Decision Letter 1]

4 Jan 2021

Dear Dr Barrés,

We are pleased to inform you that your manuscript entitled "Ablation of DNA-methyltransferase 3A in skeletal muscle does not affect energy metabolism or exercise capacity" has been editorially accepted for publication in PLOS Genetics. Congratulations!

Yours sincerely,

Yvonne Böttcher, Ph.D.

Guest Editor

PLOS Genetics

Wendy Bickmore

Section Editor: Epigenetics

PLOS Genetics

Comments from the reviewers (if applicable):

Reviewer's Responses to Questions

**Comments to the Authors:**

Reviewer #2: The authors have responded to all my comments and I have no further questions and/or comments.

Reviewer #3: The authors addressed all the comments, I have no further comments.

**Have all data underlying the figures and results presented in the manuscript been provided?**

Reviewer #2: Yes

Reviewer #3: Yes

PLOS authors have the option to publish the peer review history of their article (what does this mean?). If published, this will include your full peer review and any attached files.

Reviewer #2: No

Reviewer #3: No

**Data Deposition**

http://datadryad.org/submit?journalID=pgenetics&manu=PGENETICS-D-20-01178R1

**Press Queries**

---

## [Editor Report · Acceptance letter]

25 Jan 2021

PGENETICS-D-20-01178R1 

Ablation of DNA-methyltransferase 3A in skeletal muscle does not affect energy metabolism or exercise capacity 

Dear Dr Barrès, 

We are pleased to inform you that your manuscript entitled "Ablation of DNA-methyltransferase 3A in skeletal muscle does not affect energy metabolism or exercise capacity" has been formally accepted for publication in PLOS Genetics! Your manuscript is now with our production department and you will be notified of the publication date in due course.

With kind regards,

Alice Ellingham

PLOS Genetics

On behalf of:
